# Allogenic Faecal Microbiota Transfer Induces Immune-Related Gene Sets in the Colon Mucosa of Patients with Irritable Bowel Syndrome

**DOI:** 10.3390/biom9100586

**Published:** 2019-10-08

**Authors:** Savanne Holster, Guido J. Hooiveld, Dirk Repsilber, Willem M. de Vos, Robert J. Brummer, Julia König

**Affiliations:** 1Nutrition-Gut-Brain Interactions Research Centre, Faculty of Health and Medicine, School of Medical Sciences, Örebro University, 701 82 Örebro, Sweden; savanne.holster@oru.se (S.H.); dirk.repsilber@oru.se (D.R.); robert.brummer@oru.se (R.J.B.); 2Nutrition, Metabolism and Genomics group, Division of Human Nutrition and Health, Wageningen University, 6708 PB Wageningen, The Netherlands; guido.hooiveld@wur.nl; 3Laboratory of Microbiology, Wageningen University and Research Centre, 6708 PB Wageningen, The Netherlands; willem.devos@wur.nl; 4Human Microbiome Research Program, Faculty of Medicine, University of Helsinki, 00014 Helsinki, Finland

**Keywords:** faecal microbiota transplantation, irritable bowel syndrome, gene expression, microbiota, host-microbe interaction

## Abstract

Faecal microbiota transfer (FMT) consists of the introduction of new microbial communities into the intestine of a patient, with the aim of restoring a disturbed gut microbiota. Even though it is used as a potential treatment for various diseases, it is unknown how the host mucosa responds to FMT. This study aims to investigate the colonic mucosa gene expression response to allogenic (from a donor) or autologous (own) FMT in patients with irritable bowel syndrome (IBS). In a recently conducted randomised, double-blinded, controlled clinical study, 17 IBS patients were treated with FMT by colonoscopy. RNA was isolated from colonic biopsies collected by sigmoidoscopy at baseline, as well as two weeks and eight weeks after FMT. In patients treated with allogenic FMT, predominantly immune response-related gene sets were induced, with the strongest response two weeks after the FMT. In patients treated with autologous FMT, predominantly metabolism-related gene sets were affected. Furthermore, several microbiota genera showed correlations with immune-related gene sets, with different correlations found after allogenic compared to autologous FMT. This study shows that the microbe–host response is influenced by FMT on the mucosal gene expression level, and that there are clear differences in response to allogenic compared to autologous FMT.

## 1. Introduction

Faecal microbiota transfer (FMT) consists of the introduction of a new microbiota into the intestine of a patient, with the aim of restoring a disturbed gut microbiota. FMT has proven to be a safe and long-lasting treatment for patients with recurrent *Clostridioides difficile* infection, with cure rates of approximately 90% [1]. Faecal microbiota transfer has also shown to have positive effects in other diseases, such as ulcerative colitis (UC) and metabolic syndrome [2,3,4,5,6]. In addition, FMT seems to be a promising treatment option for irritable bowel syndrome (IBS). Irritable bowel syndrome is a common chronic gastrointestinal disorder, with an estimated worldwide prevalence of 6–18% [7,8], in which patients suffer from abdominal pain, cramps, and altered gut motility. The pathophysiology of IBS is unknown, but it is generally accepted that the microbiota–gut–brain axis plays a key role in this disorder. Aberrations along this axis include visceral hypersensitivity, altered gut microbiota, and low-grade inflammation. Three recent placebo-controlled studies have studied the effect of FMT in IBS. Johnsen et al. administered FMT by whole colonoscopy, and found more responders in the group of IBS patients receiving donor material (allogenic FMT) than in the group receiving their own stool back (autologous FMT) (*p* = 0.049) [9]. Halkjaer et al. administered FMT orally via capsules, and found greater symptom reduction after the intake of inert placebo capsules (without stool) compared to the intake of capsules containing donor faecal material [10]. We recently conducted a randomised, double-blinded, placebo-controlled clinical study, in which 17 IBS patients received FMT by colonoscopy [11]. Whereas there were no significant differences in symptoms between the allogenic and the autologous groups, possibly due to the small group size, only in the allogenic group did the symptom scores improve significantly after FMT compared to the baseline.

Even though FMT is used as a potential treatment for various diseases, there is limited knowledge on how the host mucosa responds to the introduction of a new gut microbiota. In a recent study, the impact of autologous FMT on mucosa gene expression was characterized in six antibiotic-treated, healthy individuals [12]. However, there is no research on the effect of FMT in IBS, and knowledge on mode-of-action is pivotal in order to develop a more targeted treatment approach.

The aim of the current study is to examine the effect of FMT on the gene expression in the colon mucosa of IBS patients after the infusion of allogenic (from a healthy donor) compared to autologous (own) faecal material into the colon. To the best of our knowledge, this is the first study to investigate how the host mucosa responds to the introduction of a new gut microbiota in IBS patients in a controlled fashion. 

## 2. Materials and Methods

### 2.1. Study Design

Seventeen patients with IBS were treated with faecal material from two healthy donors (allogenic FMT) or with their own faecal microbiota (autologous FMT), as described in detail in the previously published study by Holster et al. [11]. In short, two healthy donors, selected based on a high abundance of butyrate-producing bacteria in their faecal samples, were carefully screened and only included if they did not fulfill any of the strict exclusion criteria listed in Holster et al. [11]. The faecal transplant was administered by whole colonoscopy into the caecum (30 g of stool in 150 mL sterile saline). Two weeks before the FMT (baseline) as well as two and eight weeks after the FMT, the participants underwent a sigmoidoscopy, and biopsies were collected at a standardised location (20–25 cm from the anal verge, at the crossing with the *arteria iliaca communis*) from an uncleansed sigmoid. All subjects gave their written informed consent before participation in the study. The study was conducted according to the principles of the Declaration of Helsinki and its revisions, and ethical approval was obtained from the Central Ethical Review Board of Uppsala, Sweden (registration number 2013/180). The trial was registered at ClinicalTrials.gov (NCT02092402) on March 20, 2014.

### 2.2. RNA Isolation and Microarray Processing

Biopsies stored in RNAlater (Invitrogen by ThermoFisher, Waltham, MA, USA) at −80 °C were used to isolate total RNA. RNA was isolated using Qiagen RNeasy Mini Kit (Qiagen, Venlo, The Netherlands), quantified using NanoDrop, and quality was checked with the Agilent 2100 bioanalyzer (Agilent Technologies, Santa Clara, CA, USA). Samples were only included for further analyses in cases of intact bands corresponding to 18S and 28S ribosomal subunits, as well as absence of chromosomal peaks or RNA degradation products. Total RNA (100 ng) was labelled with the Whole-Transcript Sense Target Assay (Affymetrix, Life Technologies, Bleiswijk, the Netherlands; P/N 900652) and hybridized to whole genome Affymetrix Human Gene 2.1 ST arrays (Affymetrix, Santa Clara, CA, USA). Sample labelling, hybridization to chips, and image scanning were performed according to the manufacturer’s instructions. Quality control and the data analysis pipeline have been described in detail previously [13]. Briefly, normalized expression estimates of probe sets were computed by the robust multiarray analysis (RMA) algorithm [14], as implemented in the Bioconductor package AffyPLM. Probe sets were redefined using current genome information according to Dai et al. [15], based on genome annotations provided by the Entrez Gene database, which resulted in the profiling of 29,635 unique genes (custom CDF v23).

Differentially expressed probe sets (genes) were identified by using linear models (package limma) and an intensity-based, moderated *t*-statistic [16,17]. The repeated, parallel design of the study was taken into account by coding a nested model matrix. To allow for the heterogeneity in gene expression profiles in samples obtained from the same patient, a heteroskedastic model was fitted by computing weights for each sample individually that were included in the linear model [18,19]. Comparisons of gene expression data in colonic biopsies between two weeks after FMT and baseline, and eight weeks after FMT and baseline, respectively, were made for both the allogenic and autologous group (within-group comparisons). Additionally, baseline-corrected gene expression data was directly compared between the allogenic and autologous group at two and eight weeks after FMT (between-group comparisons). The *p*-values were corrected for multiple testing, according to Benjamini and Hochberg [20,21].

### 2.3. Biological Interpretation of Array Data

Changes in gene expression were related to biologically meaningful changes using gene set enrichment analysis (GSEA) [22]. It is well accepted that GSEA has multiple advantages over analyses performed on the level of individual genes [22,23,24]. Gene set enrichment analysis evaluates gene expression on the level of gene sets that are based on prior biological knowledge—e.g., published information about biochemical pathways or signal transduction routes—allowing more reproducible and interpretable analysis of gene expression data. As no gene selection step (fold change or *p*-value cut-off) is used, GSEA is an unbiased approach. A GSEA score is computed based on all genes in the gene set, which boosts the signal-to-noise ratio and allows the detection of affected biological processes that are due to only subtle changes in the expression of individual genes. This GSEA score, called a normalised enrichment score (NES), can be considered as a proxy for gene set activity. Gene sets were retrieved from the expert-curated KEGG pathway database [25]. Only gene sets comprised of more than 15 and fewer than 500 genes were taken into account. For each comparison, genes were ranked on their *t*-value, which was calculated by the moderated *t*-test. The statistical significance of GSEA results was determined using 10,000 permutations. The GSEA results were visualized for visualization and interpretation using the package clusterProfiler [26] and the Enrichment Map plugin for Cytoscape [27]. For each time point, a separate grid was made that was used for the within- and between-group comparisons.

### 2.4. Correlations between Pathway Activity and Mucosal Microbiome Data

To obtain insight into the correlations between changes in pathway activity versus the mucosa-adherent microbiota, integrative multivariate correlation analysis was performed based on data from the individual samples, using the package mixOmics [28]. Samples from the allogenic and autologous group were analysed separately, but included those obtained at baseline, as well as two and eight weeks after FMT. Unsupervised, single sample pathway scores were calculated by the gene set variation analysis (GSVA) algorithm [29], using the same pathways as used for GSEA. As input for the correlation matrix, gene sets were used that were significantly differentially regulated (*p* < 0.01) within either the allogenic or autologous group. Mucosal-adherent microbiota was analysed from colonic biopsies collected from an uncleansed sigmoid two weeks before FMT, as well as two and eight weeks after FMT, as reported previously by Holster et al. [11]. In short, microbial DNA from the mucosal samples (see Section 2.1) was isolated using repeated bead beating [30] with some adjustments, including a proteinase K incubation prior to the mechanic cell disruption, and use of a Maxwell extraction robot (Maxwell 16 Tissue LEV Total RNA Purification Kit; Promega, Madison, WI, United States). The Human Intestinal Tract Chip (HITChip), a customized Agilent microarray, was used to assess the mucosal microbiota composition, as previously described [31], with minor modifications [2]. The hybridization signals were normalized and summarized to 130 genus-like phylogenetic groups (level 2, 0.90% 16S rRNA gene sequence similarity), referred to as species and relatives [31] (See Appendix A).

The compositional, genera-level data were centered log-ratio (clr) transformed before subjected to correlation analyses [28,29]. The correlation between the two datasets (pathway activity and microbiome composition) was analysed by partial least squares (PLS) regression (canonical mode), accounting for repeated measurements obtained from the same patient. Results were visualized in clustered image maps [32].

### 2.5. Protein Isolation

Biopsies stored in AllProtect (Qiagen) at −80 °C, which were taken at the same time points as for RNA isolations, were used for protein isolation. The tissue was homogenized using the Tissuelyser (LT, Qiagen) for 3 min at 50 Hz in 250 µL RIPA buffer (Merck, Darmstadt, Germany), including 1× Protease Inhibitor Cocktail (ThermoFisher Scientific). The homogenised mixture was centrifuged for 5 min at 4 °C at 10,000 rpm (9500×g), and the supernatant was aliquoted and stored at −80 °C until further analysis.

### 2.6. Cytokine Analysis

The cytokines IFNγ, IL1-b, IL-2, IL-4, IL-6, IL-8, IL-10, IL-12p70, IL-13, and TNFα were measured in duplicates with U-PLEX Biomarker Group 1 (hu) assays (K15067L-1, Meso Scale Discovery), according to the manufacturer’s protocol. The concentrations of cytokines were normalized by the total protein, determined by a Coomassie (Bradford) protein assay kit (23200, ThermoScientific) and expressed as picogram (pg)/mg total protein. Samples with cytokine concentrations under detection level were considered to be 0 pg/mg total protein. Principal component analysis was performed on the cytokine concentrations using the prcomp-function in R with the scale set to “true”.

### 2.7. Data Availability

Array data have been submitted to the Gene Expression Omnibus under accession number GSE138297.

## 3. Results

### 3.1. Subjects

The clinical outcome and microbiota analysis of the FMT intervention has been described in detail in the previously published study of Holster et al. [11]. Seventeen IBS patients were treated with FMT, of which eight patients received faecal material from a healthy donor (allogenic FMT), and nine received their own faecal material back (autologous FMT) (for demographics of the study population, see Table 1). One of the participants dropped out due to procedure-related adverse effects, and another participant chose not to continue with the sigmoidoscopies after the FMT (both from the autologous group). In total, colonic biopsies from 15 subjects at three time points were available for analysis.

### 3.2. Individually Differentially Expressed Genes

No large impact of FMT on individual genes was found. The mean fold changes (FC) were calculated between the baseline versus two and eight weeks after FMT, respectively, in both the allogenic and the autologous group. Only genes with FC >1.5 and <−1.5 were considered as differentially expressed. In the allogenic group, thirty genes were differentially expressed, with *p* < 0.005 two weeks after FMT compared to the baseline (Appendix A), and 19 genes eight weeks after FMT compared to the baseline (Appendix A). No genes were significantly expressed after correction for multiple testing (false discovery rate, FDR < 0.05). In the autologous group, 52 differentially expressed genes with *p* < 0.005 were found two weeks after FMT, compared to the baseline (Appendix A, no significant genes after multiple testing), and 148 after eight weeks (Appendix A). After correction for multiple testing, 26 genes were significantly differentially expressed (FDR < 0.05) after eight weeks (Appendix A).

When comparing the allogenic FMT group to the autologous FMT group, 149 genes were differentially expressed two weeks after FMT with *p* < 0.005 (Appendix A). Eight weeks after FMT, 191 genes were significantly differentially expressed (*p* < 0.005; Appendix A). No genes were significantly differentially expressed after correction for multiple testing (FDR < 0.05) in both comparisons.

### 3.3. Gene Set Enrichment Analysis

#### 3.3.1. Differentially Expressed Gene Sets after Faecal Microbiota Transfer

Gene set enrichment analysis (GSEA) was performed to elucidate the biological processes that were changed upon the allogenic and autologous FMT treatments, and clear differences between the both groups were found. All pathways that were significantly differentially regulated in at least one of the six different comparisons (FDR < 0.05) are shown in Figure 1. In the allogenic group, the strongest effect was observed two weeks after FMT, where pathways were mostly upregulated compared to the baseline (Figure 1A). After eight weeks, a few of these pathways were still upregulated, while others were downregulated (Figure 1A). In the group receiving autologous FMT, fewer gene sets were significantly up- or down-regulated, and different gene sets were affected than in the group receiving allogenic FMT (Figure 1B). The direct comparison between the allogenic and the autologous group (baseline-corrected data) is shown in Figure 1C. A considerably larger number of pathways showed higher expression in the allogenic group compared to in the autologous group two weeks after FMT, and to a lesser extent, eight weeks after FMT. A different set of pathways showed lower expression in the allogenic group compared to the autologous group after eight weeks.

#### 3.3.2. Enrichment Maps

Next, enrichment maps, network-based visualizations of the GSEA results, were generated to identify common clusters of regulated pathways (Figure 2 and Figure 4). Since the focus of this study was on the differences between the treatments, we first selected the gene sets that were differentially regulated (FDR < 0.25) between the allogenic and autologous FMT after two (for Figure 2) or eight weeks (for Figure 4). These gene sets were used as input for the enrichment maps, and for better visualization, only gene sets with FDR < 0.05 in the three depicted comparisons are shown in Figure 2 and Figure 4.

##### Two Weeks after Faecal Microbiota Transfer

The enrichment map generated by comparing the response of the allogenic group to the autologous group two weeks after FMT revealed that the genes coding for immune-related pathways were higher expressed, whereas those coding for metabolic pathways were lower expressed in the allogenic group compared to the autologous FMT group (Figure 2A, see Appendix A for all gene set names). Figure 2B depicts the same enrichment map for the within-group comparison in the allogenic group after two weeks, compared to baseline. It shows that these immune-related pathways were upregulated in the allogenic FMT group, and were not affected in the autologous FMT group (Figure 2C). Figure 2C shows the within-group comparison for the autologous group at two weeks compared to baseline. Here, gene sets involved in cellular metabolism were upregulated, while they were not changed in the allogenic FMT group (Figure 2B).

The gene sets within the largest cluster of immune-related gene sets were generally representative of the activation of the adaptive immune response (allograft rejection, autoimmune thyroid disease, asthma, antigen processing and presentation, graft-versus-host disease, intestinal immune network for Immunoglobulin A production, inflammatory bowel disease, Th1 and Th2 cell differentiation, Th17 cell differentiation, type I diabetes mellitus, and viral myocarditis). The large number of connections between these gene sets imply that a common set of key genes (core enriched genes) was induced. The fold changes of these core enriched genes in both groups, as well as on individual levels, are shown in Figure 3.

In addition, Figure 3 shows which subject received FMT from which donor. The host response to the two different donors did not seem to differ notably.

##### Eight Weeks after Faecal Microbiota Transfer

Eight weeks after FMT, the allogenic group still showed an increased expression of immune-related gene sets compared to the autologous group (Figure 4A; see Appendix A for all gene set names), which seemed to be, to a large extent, attributable to a downregulation of immune-related gene sets in the autologous group eight weeks after FMT compared to baseline (Figure 4C). The upregulation of the gene sets of immune-related pathways two weeks after allogenic FMT was no longer as strong at eight weeks after FMT compared to the baseline (Figure 4B). More gene sets involved in cellular metabolism were downregulated in the allogenic group compared to the autologous group eight weeks after FMT (Figure 4A) than after two weeks (Figure 2A). The gene sets for these metabolism-related pathways were downregulated in the allogenic group and upregulated in the autologous group eight weeks after FMT, compared to the baseline (Figure 4B,C).

### 3.4. Microbiome–Gene Set Correlations

In order to generate individual gene set activity scores, gene set variation analysis (GSVA) was performed. Gene sets differentially expressed in any of the direct comparisons (allogenic vs. autologous FMT, two and eight weeks after FMT, *p* < 0.01) were correlated to the mucosa-adherent microbiota in both the allogenic FMT group (Figure 5A) and the autologous FMT group (Figure 5B). Hierarchical clustering separated the gene sets in immune-related and metabolism-related pathways. Cluster I includes the bacterial genera that were positively correlated with the immune-related pathways, while cluster II reveals the genera that were negatively correlated with the immune-related pathways in the allogenic FMT group (for individual bacterial genera names, see Table 2). In the autologous FMT group, clusters III and IV include the genera that were positively or negatively correlated to the immune-related pathways, respectively (Table 2). A large set of bacterial genus-like groups were found to correlate with the immune-related pathways. Interestingly, bacteria belonging to *Anaerostipes caccae* and *Coprococcus eutactus* were found in both FMT groups to correlate positively with these pathways. Both of these are well-known butyrate producers, with bacteria related to *A. caccae* having the property to form this intestinal signalling molecule not only from sugars, but also from acetate and lactate. Conversely, many of the bacteria that negatively correlated with the immune pathways in the autologous FMT were pathobionts belonging to the Gram-negative proteobacteria (*Helicobacter*, *Pseudonomas*, *Yersinia*, *Proteus*, and *Vibrio*) as well as various *Lactobacillus* genera that are capable of producing lactate and acetate.

### 3.5. Cytokines

In order to study whether these results were mirrored at the protein level, the concentrations of several cytokines in the biopsies, obtained at the same time points, were measured. IL-2, IL-4, IL-12p70, and IL-13 were excluded from analysis, due to too many samples with concentrations under the detection level. The levels of the detectable cytokines, tumour necrosis factor (TNF)-alpha, IL-1beta, IL-6, IL-8, IL-10, and IFN-gamma, were not significantly different between and within the groups receiving allogenic or autologous FMT (Appendix A). Appendix A show a multivariate analysis of these detectable cytokines for the allogenic group and the autologous FMT group, respectively. Also, no clear differences could be observed here.

## 4. Discussion

It is not yet known how FMT, the transfer of donor faecal material with the aim of modulating the gut microbiota of the recipient, affects the host mucosal response. In this study, the host response from IBS patients upon allogenic FMT (receiving donor material) and autologous FMT (receiving own faecal material back) was investigated. For the biological interpretation of the data, and to improve signal-to-noise ratio, gene set enrichment analysis was performed. This showed that allogenic FMT from a healthy donor evoked a different mucosal response than autologous FMT. Introduction of a new faecal microbiota ecosystem seemed to provoke a predominantly immune-related response. This response was especially strong two weeks after FMT, and was partially persisting after eight weeks. Administration of the subjects’ own faecal material induced a less profound response after two weeks, but showed an effect on both the metabolism and the immune system eight weeks after FMT compared to the baseline.

Among the cluster of gene sets found to be significantly upregulated after allogenic FMT were “allograft rejection” and “graft versus host disease”. This suggests that the host regards the new microbiota as foreign and activates defence or rejection mechanisms. Whether it is beneficial to achieve higher or lower expression of immune-related gene sets is difficult to answer. On the one hand, a strong pro-inflammatory response is generally considered as harmful; on the other hand, a stimulation of the immune response could also have beneficial effects on local and systemic immune regulation [33]. In our earlier report of the FMT trial [11], we described that symptom improvement was found in the allogenic FMT group compared to baseline, which was not the case in the autologous group. The data presented in the current study shows that the host mucosal response was different after allogenic compared to autologous FMT, which could possibly be an explanation for the symptom improvement. Although allogenic FMT seemed to evoke an immune-related response while autologous FMT did not, no serious adverse events or fever were reported after FMT [11], suggesting that the local activation of immune-related pathways did not seem to result in a systemic immune response. 

Differentially expressed gene sets were also found in the autologous group, in which patients were treated with their own faecal material. More specifically, metabolism-related gene sets were upregulated, and immune-related pathways downregulated. Our previously published results have shown that autologous FMT also has an effect on the faecal and mucosal microbiota, possibly due to the prior cleansing of the bowel or the handling of the faecal material during preparation for the transplant. These changes in gut microbiota composition could be an explanation for an altered gene expression response upon autologous FMT.

Not much is known about the effect of FMT on the mucosal host response. In vitro, it has been shown that microbiota from a healthy donor alters the expression of numerous host genes in primary human colonic epithelial cells [34]. A brief report showed that after repeated administration of FMT in three paediatric UC patients, gene expression profiles in the colon mucosa had changed. The genes downregulated after FMT were involved in leukocyte activation and mitotic cell cycle progression processes [35]. However, it needs to be noted that in this study, baseline biopsies were collected from a prepared bowel, while follow-up samples from two of the patients were collected from an uncleansed bowel. In our study, all biopsies were collected from an uncleansed bowel at a standardised location in the sigmoid.

While there is a lack of studies with regards to FMT, specific strains of the gut microbiota have previously been shown to be able to modulate the gene expression of the host. After oral ingestion of the commensal strain *Lactobacillus plantarum*, gene expression profiles in the duodenal mucosa of healthy adults were altered, with several immune-related pathways affected [36]. Different probiotic species have been shown to have different effects on duodenal gene expression [37], and even strain-dependent effects have been described [38]. For example, the *L. plantarum* strain TIFN101 was shown to upregulate gene expression pathways involved in T and B cell function, as well as antigen presentation, in the duodenal mucosa of healthy subjects. However, in the same study other *L. plantarum strains*, CIP48 and WCFS1, were shown to have a tendency to downregulate pathways involving antigen presentation processes in the mucosa [38]. In our study, we identified several bacterial genera that correlated either positively or negatively with immune-related gene sets, with different correlations found after allogenic compared to autologous FMT. This substantiates again that FMT not only alters the intraluminal intestinal ecosystem, but also specifically affects microbe–host interactions. 

Even though the gut microbiota is a complex ecosystem, and we only had a limited study population, we saw a rather consistent effect of FMT on the gene expression in the colonic mucosa. An advantage of our study was that biopsies were collected at two time points after FMT, which confirms the robustness of the results. In addition, we used a controlled study design. However, in future studies, a separate control group that only receives bowel cleansing, as well as a different choice of placebo, could give valuable insights. A limitation of this study was that blood samples were not collected, and therefore only mucosal cytokine production was studied. No clear differences were observed between cytokine levels after allogenic and autologous FMT, which could indicate that the protein production in biopsies were not affected by FMT, or that changes were not measurable due to low detection levels.

## 5. Conclusions

In conclusion, we can show that the gut microbiota–host response is affected by FMT on the mucosal gene expression level, and that there are clear differences in response to allogenic compared to autologous FMT. How these effects contribute to a successful outcome in FMT therapy, especially with regard to precision medicine, needs to be further elucidated in future studies.

## Figures and Tables

**Figure 1 biomolecules-09-00586-f001:**
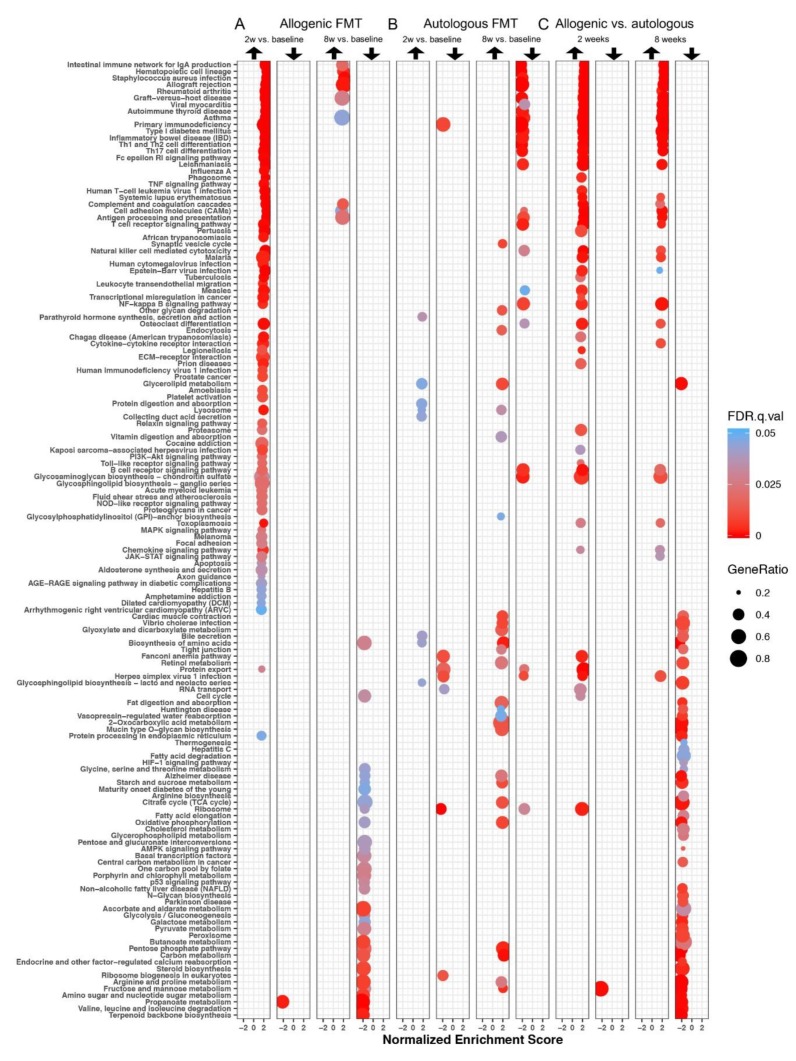
Differentially expressed gene sets after faecal microbiota transfer (FMT). (**A**) Allogenic FMT (within-group comparisons). (**B**) Autologous FMT (within-group comparisons). (**C**) Allogenic versus autologous FMT (two and eight weeks after FMT, baseline-corrected). Gene set enrichment analysis (GSEA) was performed. Each row depicts a gene set that was significantly differentially regulated in one of the six comparisons (false discovery rate, FDR < 0.05), with the corresponding normalised enrichment score (NES) value on the *y*-axis. The colour of the dot indicates the statistically significant FDR value of the corresponding gene set in that specific comparison. The size of the dot reflects the gene ratio, which represents the number of enriched genes in the gene set.

**Figure 2 biomolecules-09-00586-f002:**
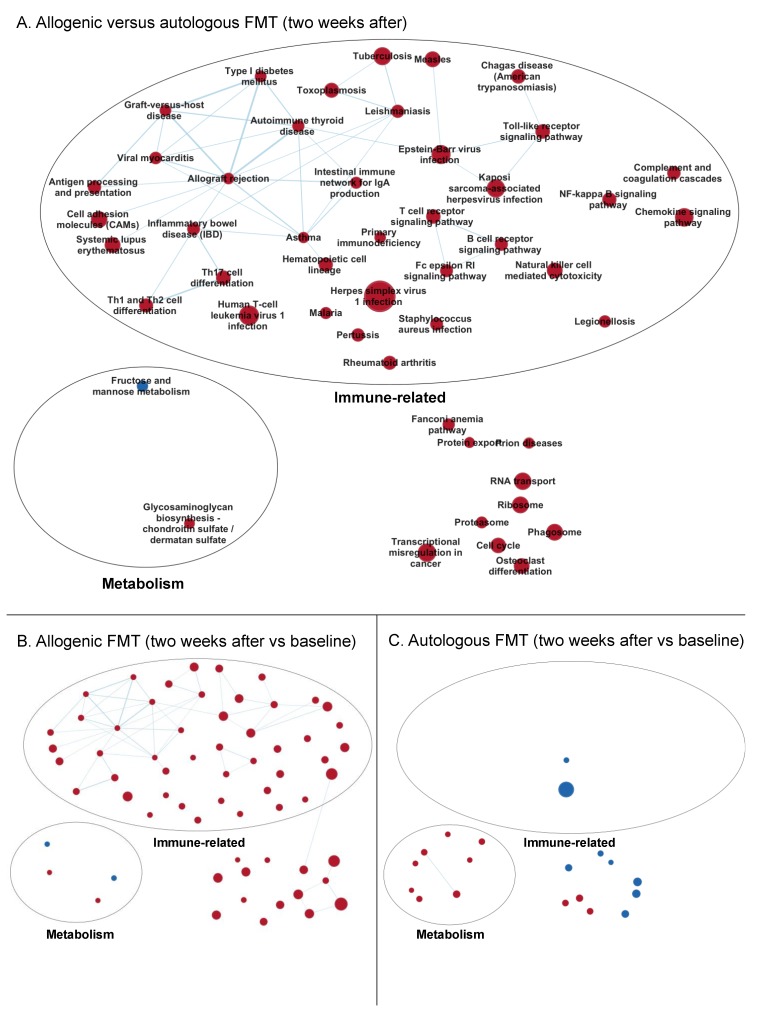
Enrichment maps of the gene set changes two weeks after FMT. (**A**) Allogenic versus autologous FMT (baseline-corrected). (**B**) Allogenic FMT (within-group comparison). (**C**) Autologous FMT (within-group comparison). Nodes represent KEGG gene sets, and the edges between the nodes represent their similarity. Red nodes indicate enriched (increased expression of) gene sets, and blue nodes indicate suppressed (decreased expression of) gene sets. Node size represents the number of genes in the gene set, and the thickness of the edges indicates the degree of overlap between the two connected gene sets (nodes). The gene sets are manually grouped according to their biological functions among these gene sets. See Appendix A for the enrichment map including all the names of the gene sets. Gene sets that were differentially regulated (FDR < 0.25) between the allogenic and autologous FMT after two weeks were used as input for the enrichment map, and for better visualization, only the gene sets with FDR < 0.05 from the three depicted comparisons are shown.

**Figure 3 biomolecules-09-00586-f003:**
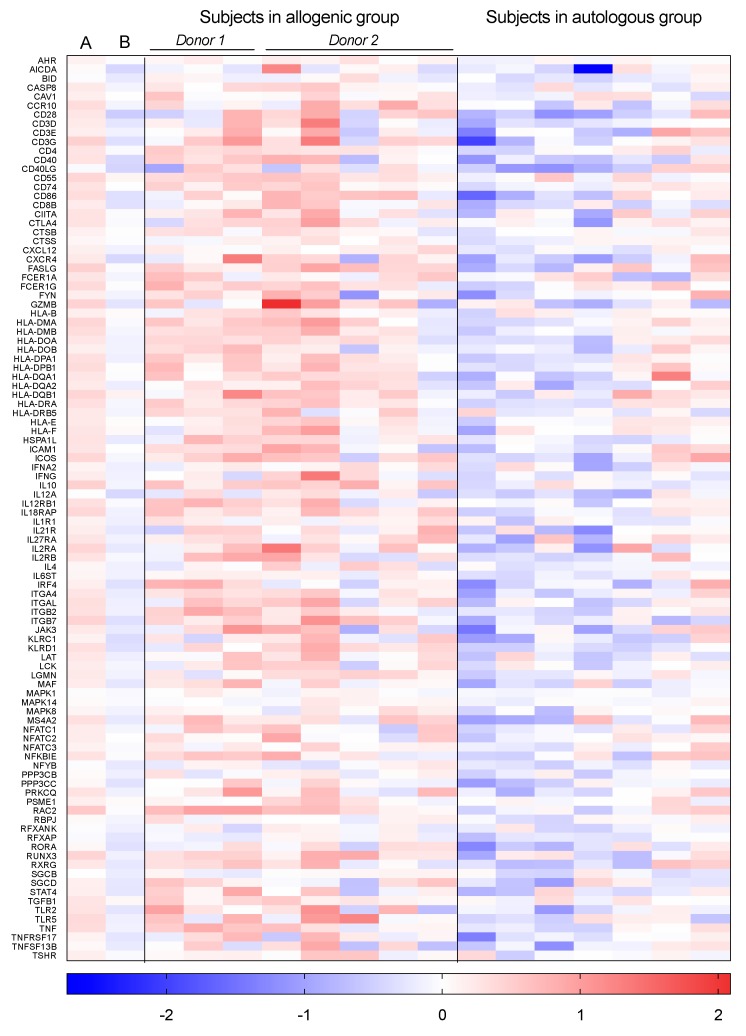
Core enriched genes shared by the gene sets in the largest network of the enrichment map two weeks after FMT. The log2 fold changes of the genes commonly induced in the gene sets within the largest cluster of immune-related gene sets (Figure 2A) are shown. The first two columns show the log2 fold changes after allogenic FMT compared to baseline (**A**) and after autologous FMT compared to baseline (**B**), respectively. The two last panels show the log2 fold changes per subject after allogenic and autologous FMT, respectively. The lines above the subjects receiving allogenic FMT indicate which subject received FMT from which donor.

**Figure 4 biomolecules-09-00586-f004:**
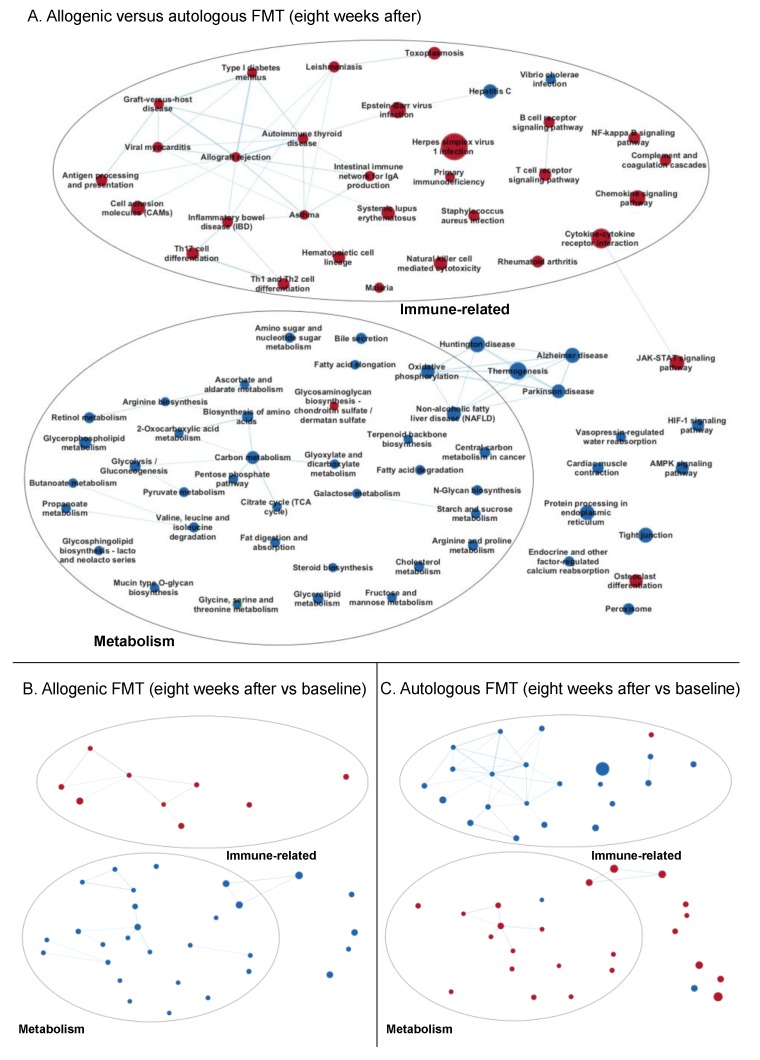
Enrichment maps of gene sets changed eight weeks after FMT. (**A**) Allogenic versus autologous FMT (baseline-corrected). (**B**) Allogenic FMT (within-group comparison). (**C**) Autologous FMT (within-group comparison). Nodes represent KEGG gene sets, and the edges between the nodes represent their similarity. Red nodes indicate enriched (increased expression of) gene sets, and blue nodes indicate suppressed (decreased expression of) gene sets. Node size represents the number of genes in the gene set, and the thickness of the edges indicates the degree of overlap between the two connected gene sets (nodes). The gene sets are manually grouped according to their biological functions among these gene sets. See Appendix A for the enrichment map including the names of all the gene sets. Gene sets that were differentially regulated (FDR < 0.25) between the allogenic and autologous FMT after eight weeks were used as input for the enrichment map, and for better visualization, only the gene sets among those with FDR < 0.05 in the three depicted comparisons are shown.

**Figure 5 biomolecules-09-00586-f005:**
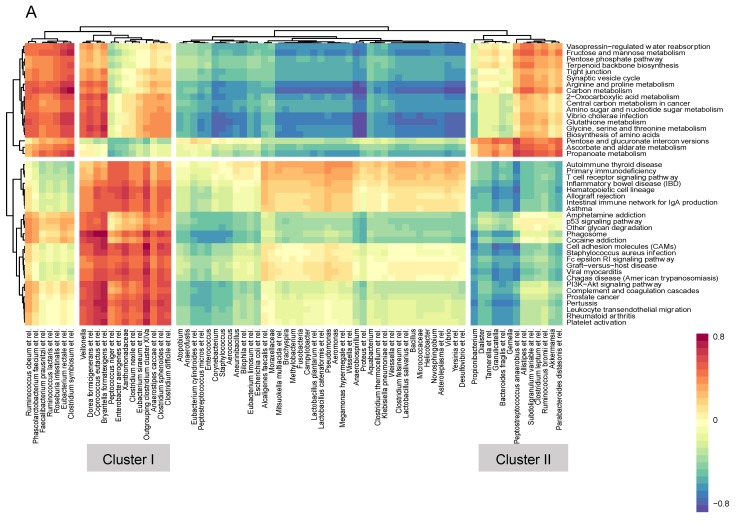
Heat map of correlations between mucosa-adherent microbiota and gene sets affected by FMT. (**A**) Allogenic FMT. (**B**) Autologous FMT. Gene sets differentially expressed in any of the direct comparisons (allogenic vs. autologous FMT, two and eight weeks after FMT, *p* < 0.01) were included. Correlations with correlation coefficient (*r)* > 0.5 are shown. GSVA: gene set variation analysis.

**Table 1 biomolecules-09-00586-t001:** Baseline characteristics of irritable bowel syndrome (IBS) patients included in this study.

	Allogenic (*n* = 8)	Autologous (*n* = 7)	*p*-Value
Age, median (IQR)	34 (27–42)	38 (32–45)	0.42
Sex, m/f	5/3	3/4	0.62
BMI (kg/m^2^), median (IQR)	20.9 (20.2–25.1)	23.8 (20.5–24.7)	0.94
Classification, IBS-D/IBS-C/IBS-M	5/1/2	4/2/1	1.0/0.57/1.0
Post-infectious IBS	4	3	1.0
Disease duration: unknown/1–5 y/5 y	0/4/4	1/3/3	0.47/1.0/1.0
Concomitant medication	7	5	0.57
Gut-related medication	3	3	1.0
Laxatives	1	2 *	1.0
Anti-diarrhoeal	1	1	1.0
Anti-spasmodic	1	1	1.0
Antidepressants	5	2	0.31
SSRIs	5	0	0.03
NaSSAs	0	1	0.47
SSNRIs	0	1#	0.47
TCAs	0	1#	0.47

IQR: interquartile range, BMI: body mass index, D: diarrhoea, C: constipation, M: mixed classification, SSRI: serotonin-reuptake inhibitor, NaSSA: noradrenergic and specific serotonergic antidepressant, SSNRI: selective serotonin–noradrenalin-reuptake inhibitor, TCA: tricyclic antidepressant. * One participant took two different types of laxatives. #One participant took both SSNRI and TCA. This table has been modified from Holster et al. [11].

**Table 2 biomolecules-09-00586-t002:** Bacterial genera positively (+) or negatively (−) correlating to immune-related gene sets affected by FMT.

Allogenic FMT		Autologous FMT	
Cluster I (+)	Cluster II (−)	Cluster III (+)	Cluster IV (−)
*Peptococcus niger et rel.*	*Propionibacterium*	*Anaerovorax odorimutans et rel.*	*Aneurinibacillus*
*Enterobacter aerogenes et rel.*	*Bacteroides fragilis et rel.*	*Phascolarctobacterium faecium et rel.*	*Bilophila et rel.*
*Xanthomonadaceae*	*Gemella*	*Clostridium orbiscindens et rel.*	*Lactobacillus plantarum et rel.*
*Clostridium nexile et rel.*	*Granulicatella*	*Sporobacter termitidis et rel.*	*Clostridium (sensu stricto)*
*Outgrouping clostridium cluster XIVa*	*Dialister*	*Oscillospira guillermondii et rel.*	*Uncultured Chroococcales*
*Eubacterium siraeum et rel.*	*Tannerella et rel.*	*Oxalobacter formigenes et rel.*	*Fusobacteria*
*Clostridium sphenoides et rel.*	*Peptostreptococcus anaerobius et rel.*	*Faecalibacterium prausnitzii et rel.*	*Pseudomonas*
*Clostridium difficile et rel.*	*Akkermansia*	*Ruminococcus bromii et rel.*	*Uncultured Mollicutes*
*Anaerostipes caccae et rel.*	*Parabacteroides distasonis et rel.*	*Clostridium cellulosi et rel.*	*Novosphingobium*
*Coprococcus eutactus et rel.*	*Clostridium leptum et rel.*	*Ruminococcus lactaris et rel.*	*Wissella et rel.*
*Bryantella formatexigens et rel.*	*Ruminococcus bromii et rel.*	*Ruminococcus callidus et rel.*	*Yersinia et rel.*
*Veillonella*	*Allistipes et rel.*	*Coprococcus eutactus et rel.*	*Peptococcus niger et rel.*
*Dorea formicigenerans et rel.*	*Subdoligranulum variable at rel.*	*Eubacterium hallii et rel.*	*Proteus et rel.*
		*Papillibacter cinnamivorans et rel.*	*Vibrio*
		*Bacteroides uniformis et rel.*	*Moraxellaceae*
		*Clostridium sphenoides et rel.*	*Prevotella ruminicola et rel.*
		*Outgrouping clostridium cluster XIVa*	*Lactobacillus catenaformis et rel.*
		*Ruminococcus obeum et rel.*	*Micrococcaceae*
		*Clostridium symbiosum et rel.*	*Helicobacter*
		*Anaerostipes caccae et rel.*	*Clostridium felsineum et rel.*
			*Methylobacterium*
			*Lactobacillus salivarius et rel.*
			*Aeromonas*
			*Desulfovibrio et rel.*
			*Clostridium thermocellum et rel.*
			*Campylobacter*
			*Asteroleplasma et rel.*
			*Bacillus*
			*Brachyspira*
			*Actinomycetaceae*
			*Aerococcus*
			*Anaerobiospirillum*
			*Aquabacterium*
			*Atopobium*
			*Bulleidia moorei et rel.*
			*Catenibacterium mitsuokai et rel.*
			*Corynebacterium*
			*Megasphaera elsdenii et rel.*
			*Mitsuokella multiacida et rel.*
			*Peptostreptococcus anaerobius et rel.*
			*Xanthomonadaceae*

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
