# Peer review of "Allogenic Faecal Microbiota Transfer Induces Immune-Related Gene Sets in the Colon Mucosa of Patients with Irritable Bowel Syndrome"

_biomolecules, 2019, doi:10.3390/biom9100586_

Round 1
Reviewer 1 Report
Minor revisions that need to be addressed in order to give a better understanding and visualization of significant data:
Results
3.1. Subjects
Description of the study cohort under investigation should be improved. The authors should consider to indicate patient’s age at sample collection. Also, it could be useful to add other patient’s clinical data, such as medication/antibiotics in use at the time of sample collection (if any), as these can have an influence on downstream microbiota analysis.
3.2. Individually differentially expressed genes
Supplementary Tables: Since the identified differentially expressed genes are not significantly expressed after correction for multiple testing (FDR > 0.05), the authors should consider to change the title of the Table S1 and S2 legends, and to add the FDR values to the tables. Same as for Tables S3, S4, S6 and S7.
The authors should consider to move Table S5 in the manuscript, and a column with the corresponding p values (before multiple comparison) should be added.
3.3. Gene set enrichment analysis and 3.4. Microbiome‐gene sets correlations
It is confusing how the authors selected data for their analysis (lanes 2010-2014).
In general, the authors should consider to move results with no significant FDR values in the supplementary material, and threshold of significance for the FDR value should be clearly indicated. It is unclear which are the data with significant p-values but with no significant FDR values.
Authors should consider revising this part as it is not easily understandable for the reader.
Figure 1, 2 4, and 5, as well as Figure S1 and S2 quality should be improved, as it is not possible to read gene set names.
3.5. Cytokines
The authors should consider moving Figure 6 to the supplementary material. Also, please clarify if there is a mistake in Figure 6G and Figure 6H as the PCA appear empty.
Author Response
POINT TO POINT RESPONSE TO REVIEWER 1:
Minor revisions that need to be addressed in order to give a better understanding and visualization of significant data:
Results
3.1. Subjects
Description of the study cohort under investigation should be improved. The authors should consider to indicate patient’s age at sample collection. Also, it could be useful to add other patient’s clinical data, such as medication/antibiotics in use at the time of sample collection (if any), as these can have an influence on downstream microbiota analysis.
Thank you very much for reviewing our manuscript! We have added a table with this information (new Table 1 on page 5).
3.2. Individually differentially expressed genes
Supplementary Tables: Since the identified differentially expressed genes are not significantly expressed after correction for multiple testing (FDR > 0.05), the authors should consider to change the title of the Table S1 and S2 legends, and to add the FDR values to the tables. Same as for Tables S3, S4, S6 and S7.
Thank you for this comment. We have changed the titles and legends of table S1-S7 and added the FDR values and p-values.
The authors should consider to move Table S5 in the manuscript, and a column with the corresponding p values (before multiple comparison) should be added.
Thank you for this comment! We added the corresponding p values to Table S5. We would prefer not to move Table S5 to the manuscript as we do not focus on the individually differentially expressed genes in this specific comparison. We could however do that if you prefer. We also noticed that there was still some not updated data in Supplemental Table 5, and have now changed this. The respective information in the text has been correct from the beginning.
3.3. Gene set enrichment analysis and 3.4. Microbiome‐gene sets correlations
It is confusing how the authors selected data for their analysis (lanes 2010-2014).
We apologise for the confusion and have now changed the text in line 255 (in section ‘3.3.2 Enrichment maps').
In general, the authors should consider to move results with no significant FDR values in the supplementary material, and threshold of significance for the FDR value should be clearly indicated. It is unclear which are the data with significant p-values but with no significant FDR values.
Thank you for this comment. In order to create clarity, we have made Figure 1 again with a a cut-off value of FDR<0.05. For the enrichment maps (Figure 2 and 4), gene sets that were differentially regulated (FDR<0.25) between the allogenic and autologous FMT after two weeks (Figure 2) and eight weeks (Figure 8) were used as input for the enrichment map and, for better visualization, only the gene sets among those with FDR<0.05 in the three depicted comparisons are shown.
Authors should consider revising this part as it is not easily understandable for the reader.
We hope that it is easier to understand now after our revisions.
Figure 1, 2 4, and 5, as well as Figure S1 and S2 quality should be improved, as it is not possible to read gene set names.
We apologise for the inconvenience. The names of the pathways are increased in Figure 1, 2, 4, 5, S1 and S2. When zooming into S1b,c and S2b,c the gene set names should be readable. We will double-check with the type-setters that the quality will be good enough in the final version. In order to increase the visibility of the clusters in figure 5, we used a specific R code to identify the current clusters instead of manual identification which we had applied in the first manuscript. As a result, 12 additional bacteria genera were identified in cluster IV, which we also added to Table 2.
3.5. Cytokines
The authors should consider moving Figure 6 to the supplementary material. Also, please clarify if there is a mistake in Figure 6G and Figure 6H as the PCA appear empty.
We have now added Figure 6 to the Supplementary figures (now Figure S3). We apologise for the missing PCA plots in this figure and hope it is fixed now (improved conversion to pdf).
Reviewer 2 Report
Manuscript: biomolecules-574690
Article type: research article
Title: Allogenic faecal microbiota transfer induces immune-related gene sets in the colon mucosa of patients with irritable bowel syndrome
Authors: Savanne Holster, Guido J. Hooiveld, Dirk Repsilber, Willem M. de Vos, Robert J. Brummer, Julia Konig
Journal: Biomolecules
General comments:
The submitted article describes changes in gene profile of the colon after the FMT in patients with IBS. Actually, it is the continuation of the previous paper of this group (Holster et al. 2019).
Holster, S., Lindqvist, C.M., Repsilber, D., Salonen, A., de Vos, W.M., Konig, J., Brummer, R.J., 2019. The Effect of Allogenic Versus Autologous Fecal Microbiota Transfer on Symptoms, Visceral Perception and Fecal and Mucosal Microbiota in Irritable Bowel Syndrome: A Randomized Controlled Study. Clin Transl Gastroenterol. 10(4), e00034. doi: 10.14309/ctg.0000000000000034.
Authors used data from this previous study and correlated them with the new data from microarray analysis. They found that predominantly immune response-related genes were induced. However, It is not surprising that immune system is responding to the allogenic transplanted microbiota by its activation.
The presented manuscript is very well written, it presents new data obtained mainly by bioinformatics technology and it is conceptually clear. However, in the previous paper, 2 different donors of stools were used for the FMT, but in the present study, authors claim that “faecal material from a healthy donor”, (line 70) was used. Since they used the microbiome data from the experiments with 2 donors, whose microbiota surely differ significantly, the analysis and correlation should be done for each donor separately. The elucidation of the use of donor stools is necessary.
Major comments:
1) There is no clear the origin of donors stools.
2) There are no data about the preparation of microbial samples for sequencing. Authors refer to previous study, but I consider these information be available in this manuscript as helpful. Moreover, there is no data link to the obtained sequencing data, which should be available for public.
3) Fig. 1. There is no explanation, how differ 1st and 3rd , respectively 2nd and 4th columns. I suppose they represent 2weeks and 8weeks, but it should be indicated in the picture.
4) Figure 6. The figure in the manuscript is empty!
5) The measurement of mucosal cytokines was not very efficient, the measurement of the circulating cytokines in serum would provide more conclusive results. But this limitation was mentioned by authors in the manuscript.
6) The discussion part includes largely the results from the previous study Holster et al. 2019. The discussion of new results would be more appropriate.
Minor comments:
l.384, 396, 402 – latine names should be in italic.
Author Response
POINT TO POINT RESPONSE TO REVIEWER 2:
General comments
The submitted article describes changes in gene profile of the colon after the FMT in patients with IBS. Actually, it is the continuation of the previous paper of this group (Holster et al. 2019).
Authors used data from this previous study and correlated them with the new data from microarray analysis. They found that predominantly immune response-related genes were induced. However, It is not surprising that immune system is responding to the allogenic transplanted microbiota by its activation.
The presented manuscript is very well written, it presents new data obtained mainly by bioinformatics technology and it is conceptually clear. However, in the previous paper, 2 different donors of stools were used for the FMT, but in the present study, authors claim that “faecal material from a healthy donor”, (line 70) was used. Since they used the microbiome data from the experiments with 2 donors, whose microbiota surely differ significantly, the analysis and correlation should be done for each donor separately. The elucidation of the use of donor stools is necessary.
Thank you for your positive reply and this valuable comment! We have now corrected the information in line 70 (now line 69) to two donors. Unfortunately, a separate statistical analysis for each donor would lack power, however, we have done a preliminary re-analysis of the dotplot in Figure 1, see attachment. The dotplot below shows the gene sets enriched two weeks after FMT from donor 1 (n=3, column 3 and 4), from donor 2 (n=5, column 5) and from all patient receiving allogenic FMT together (column 1 and 2). The host response to FMT from donor 1 is not very different from the host response to FMT from donor 2. Especially the immune-related gene sets show the same pattern.
Additionally, we have added the donor-recipient relation to the heatmap in Figure 3 and this text in line 280: In addition, Figure 3 shows which subject received FMT from which donor. The host response to the two different donors did not seem to differ notably.
Major comments:
1) There is no clear the origin of donors stools.
Thank you for pointing this out. We have now added more information to line 71 (in section ‘2.1. Study design’)
2) There are no data about the preparation of microbial samples for sequencing. Authors refer to previous study, but I consider these information be available in this manuscript as helpful. Moreover, there is no data link to the obtained sequencing data, which should be available for public.
Thank you for this comment! We have added the information to line 137 (in section ‘2.4. Correlations between pathway activity and mucosal microbiome data’) and added the HitChip data as a supplemental file.
3) Fig. 1. There is no explanation, how differ 1st and 3rd , respectively 2nd and 4th columns. I suppose they represent 2weeks and 8weeks, but it should be indicated in the picture.
Thank you for pointing out this mistake. We have corrected it in Figure 1.
4) Figure 6. The figure in the manuscript is empty!
We apologise for the missing PCA plots in this figure and hope it is fixed now (problems with pdf conversion). As wished by the other reviewer, we have moved this figure to the supplemental figures (now Figure S3).
5) The measurement of mucosal cytokines was not very efficient, the measurement of the circulating cytokines in serum would provide more conclusive results. But this limitation was mentioned by authors in the manuscript.
True, unfortunately we did not collect blood in this study.
6) The discussion part includes largely the results from the previous study Holster et al. 2019. The discussion of new results would be more appropriate.
Thank you for this comment. We have tried to mostly discuss the new results, and to include the previous results only if necessary. Paragraph 1, 4, and 5 only include discussion of the new results. Only in paragraph 2 and 3 we related the new results to relevant previous results on symptoms and the effect of bowel cleansing. We hope that this is acceptable for the reviewer.
Minor comments:
l.384, 396, 402 – latine names should be in italic.
Thank you for indicating these mistakes, we have changed the formatting in lines 428, 440 and 445.
